# Enhancing Physician Flexibility: Prompt-Guided Multi-class Pathological Segmentation for Diverse Outcomes

Can Cui
*Computer Science*
*Vanderbilt University*
Nashville, USA
can.cui.1@vanderbilt.edu

Ruining Deng
*Computer Science*
*Vanderbilt University*
Nashville, USA
r.deng@vanderbilt.edu

Junlin Guo
*Computer Science*
*Vanderbilt University*
Nashville, USA
junlin.guo@vanderbilt.edu

Quan Liu
*Computer Science*
*Vanderbilt University*
Nashville, USA
quan.liu@vanderbilt.edu

Tianyuan Yao
*Computer Science*
*Vanderbilt University*
Nashville, USA
tianyuan.yao@vanderbilt.edu

Haichun Yang
*Computer Science*
*Vanderbilt University*
Nashville, USA
haichun.yang@vumc.org

Yuankai Huo
*Computer Science*
*Vanderbilt University*
Nashville, USA
yuankai.huo@vanderbilt.edu

*Abstract*—The Vision Foundation Model has recently gained attention in medical image analysis. Its zero-shot learning capabilities accelerate AI deployment and enhance the generalizability of clinical applications. However, segmenting pathological images presents a special focus on the flexibility of segmentation targets. For instance, a single click on a Whole Slide Image (WSI) could signify a cell, a functional unit, or layers, adding layers of complexity to the segmentation tasks. Current models primarily predict potential outcomes but lack the flexibility needed for physician input. In this paper, we explore the potential of enhancing segmentation model flexibility by introducing various task prompts through a Large Language Model (LLM), compared to traditional task ID tokens. Our contribution is in four-fold: (1) we construct a computational-efficient pipeline that uses finetuned language prompts to guide flexible multi-class segmentation; (2) We compare segmentation performance with fixed prompts against free-text; (3) We design a multi-task kidney pathology segmentation dataset and the corresponding various free-text prompts; and (4) We evaluate our approach on the kidney pathology dataset, assessing its capacity to new cases during inference.

*Index Terms*—Medical image segmentation, Renal pathology, Visual-language model

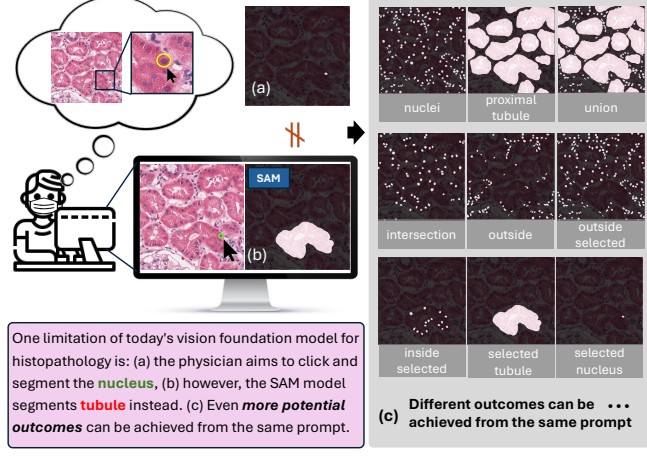

Fig. 1. **Problem definition**: For pathology images, the small, diverse structures and their complex relationships demand higher flexibility in image segmentation, which current segmentation methods may not meet. Sometimes, the segmentation target is ambiguous without the language prompt.

## I. INTRODUCTION

In the field of pathology, accurate image analysis of various tissue regions, functional units, and individual cells is crucial for disease diagnosis, treatment planning, and research exploration. Precise and reliable image analysis aids pathologists in identifying abnormalities, understanding disease progression, and formulating effective treatment strategies. With the rapid development of deep learning technologies, numerous multi-class segmentation models have been proposed to enhance pathology image analysis. These models aim to segment images into multiple predefined categories, each representing a different type of tissue or cellular structure.

Most of the existing segmentation models are based on traditional multi-class segmentation approaches, which rely on defining a fixed number of classes in advance [1], [2], [3]. Such models often face limitations due to their high memory consumption, particularly when dealing with a large number of classes. This is because traditional models typically utilize multiple channels or multi-head architectures to handle different classes, leading to increased computational demands and resource usage.

In clinical practice, the needs of pathologists often differ from the capabilities provided by these traditional models. Instead of requiring the segmentation of entire regions in

pathology images, pathologists might be more interested in annotating specific units within certain regions for detailed statistical analysis. This targeted approach allows for a more nuanced understanding of the tissue and cellular structures relevant to particular diseases or research questions.

Recently, foundational models that incorporate spatial prompts, such as bounding boxes and points, have been introduced to guide the segmentation process [4], [5], [6]. They offer a more flexible and more interactive approach to image segmentation. However, these spatial prompts can sometimes be unclear or ambiguous, particularly in the context of pathology within medical imaging. Pathology images often contain small, diverse structures with complex relationships, demanding a higher degree of precision and adaptability in segmentation methods.

For instance, consider the scenario depicted in Figure 1. When a single point is used as a prompt for segmentation, it may be ambiguous whether the target for segmentation is the cell or the tubule centered by the point. The complexity increases when pathologists are interested in more specific tasks, such as segmenting all cells within a particular tubule identified by a single point. Such intricate requirements highlight the need for advanced segmentation techniques that can provide higher flexibility and accuracy to meet the specific demands of medical imaging in pathology.

In such cases, language prompts provide the additional help of clarity to describe the target more accurately. For example, in Figure 1, when combined with a language prompt to identify the object of nuclei or tubule, the pathologist can provide a flexible yet clear request, Along with the simple spatial prompts, this approach ensures precise and targeted segmentation, enhancing the accuracy and usefulness of the analysis.

The development of large language models (LLMs) is booming, leading to significant advancements in natural language processing and understanding [7]. Models such as GPT-4 [8], BERT [9], and Llama [10] have revolutionized applications like machine translation, sentiment analysis, text summarization, and conversational agents. Their ability to generate human-like text and understand complex language nuances has opened new possibilities for enhancing human-computer interactions. Inspired by these advancements, we propose leveraging LLMs to guide segmentation using language, significantly enhancing the flexibility of traditional segmentation models.

While previous models like SEEM [11] and LiSA [12] have proposed language-guided segmentation in natural image domains. Both of them were trained by a large number of paired natural image and text data but the trained network has not been successful in pathology image segmentation due to the specific expertise required. Our goal is to develop a more efficient pipeline that utilizes the pre-trained weights from the foundation model and can be fine-tuned on relatively small datasets, making them more accessible and affordable for medical image analysis. By integrating language prompts with spatial cues, we aim to improve the interpretability and usability of segmentation models, leading to more accurate diagnoses,

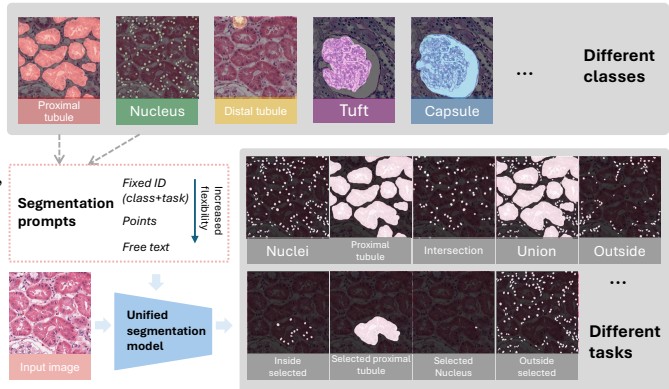

Fig. 2. **Idea of our work**: The addition of free text provides clarity for accurately describing the segmentation target compared with using the spatial annotation only. We simulate a multi-class and multi-task dataset of kidney pathology in this work. **Multiple classes of units** are essential in renal pathology image analysis, such as the proximal tubule, distal tubule, tuft, capsule, nuclei, etc. For each class, there can be **different segmentation tasks**. This figure shows the unit classes and tasks we prepared for our work. Our approach investigates the efficacy of controlling segmentation models through natural language prompts and point-based methods, highlighting the enhanced flexibility of these prompts in contrast to conventional fixed task IDs.

better treatment planning, and insightful research findings.

The contributions of this work can be summarized as follows:

- We introduce a pipeline that utilizes EfficientSAM [5] as the backbone. This pipeline incorporates free-text embeddings from TinyLlama-1.1B (fine-tuned by LoRA) and spatial embeddings of points as prompts to guide multi-class and multi-task kidney pathology image segmentation.
- We conduct a comparison among the use of free-text prompts and two strategies of fixed ID embeddings in guiding the segmentation process.
- We design a multi-task and multi-class segmentation dataset and the corresponding various free-text prompts by using a public multi-class kidney pathology segmentation dataset.
- We evaluate our approach to the kidney pathology dataset, assessing its ability to handle new segmentation cases during inference.

## II. RELATED WORK

### A. Multi-class pathology segmentation

The anatomical quantification of pathology is crucial for many clinical applications and research projects in this field. Most existing work on multi-class segmentation in pathology images uses traditional methods with multi-channel or multi-head outputs. However, these methods are limited to a fixed number of classes, requiring changes in network structure when the number of classes changes. Additionally, the increase in output channels results in higher memory consumption as the number of classes increases.

Recently, a unified multi-scale, multi-class segmentation model called Omniseg [13] was introduced and applied to renal pathology. Omniseg utilizes embeddings of task IDs to specify the class for segmentation and employs a dynamic head to control the model, enabling it to learn task-dependent weights of certain layers and adapt to different class segmentations. This approach reduces the need for multi-channel or multi-head outputs to a unified 2-channel output for multi-class segmentation. The input task ID determines what the foreground and background are in each task.

More recently, in their extension work HATs [14], a vit-based backbone using the efficientSAM [5] with the dynamic head was proposed. However, this approach is still limited to a predefined number of tasks and has not been evaluated on unseen cases during the inference phase.

Inspired by Omniseg and HATs, we aim to extend the flexibility of the task ID to free-text prompts. This enhancement will enable the model to handle a broader range of complex segmentation cases, dynamically adapting to various tasks. Such a setting enhances the model's practicality in clinical settings, facilitating precise and efficient anatomical quantification in pathology.

### B. Language-guided segmentation

The Segment Anything Model (SAM) [4] was introduced as a foundational segmentation model capable of zero-shot segmentation using spatial annotations such as points and boxes as the prompts. More recently, EfficientSAM was developed based on the distillation of a trained SAM model, achieving competitive performance with fewer parameters. In the medical field, models like MedSAM [6] have adapted SAM for medical applications.

In the language domain, models such as BERT [9] and LLaMA [10] have demonstrated remarkable zero-shot capabilities. These pre-trained language models encode natural language, serving as interfaces to integrate with other models, greatly enhancing design and functional flexibility. For instance, CLIP aligns language with images, while LiSA uses language inputs for reasoning, incorporating language tokens into image segmentation networks to present question-answer tasks as image segmentation.

Moreover, approaches like SEEM [11] and LiSA [12] have implemented segmentation guided by language prompts within the natural image domain. SEEM integrates spatial sparse annotation, language, and image tokens into transformer networks, utilizing intra- and inter-modal self-attention to perform tasks such as image editing based on spatial annotations. LiSA processes language prompts using a fine-tuned vision-language model and incorporates the predicted language token into pre-trained segmentation models like SAM to achieve final segmentation results.

However, these Vision-Language Model (VLM) efforts have primarily focused on natural image segmentation, lacking abundant medical information, particularly specialized terminology and medical images, in large-scale model training datasets.

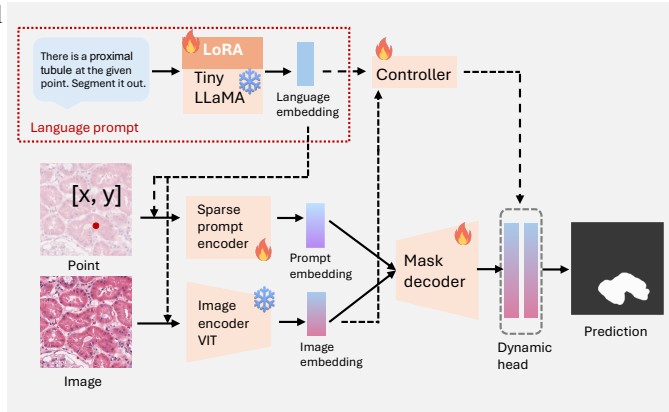

Fig. 3. **Proposed pipeline**: This figure presents the pipeline using free text and points as segmentation prompts. The lower part illustrates the segmentation backbone, while the upper part shows how the embeddings of free-text prompts are generated and integrated into the segmentation backbone in three stages. The trainable blocks and frozen blocks are highlighted with fire and ice icons, respectively.

Inspired by LiSA's structure, our model incorporates predicted language tokens from the language modality as prompts for feeding into pre-trained segmentation models. We have adapted this approach using lightweight models suitable for small medical image datasets, while also leveraging spatial sparse annotations to guide the segmentation process.

## III. METHODS

This study aims to enhance multi-class, multi-task renal pathology segmentation by integrating language prompts and sparse spatial annotation prompts. We hypothesize that combining spatial information from sparse annotations (such as points, scribbles, and bounding boxes) with semantic descriptions from language provides both flexibility and precise guidance for segmentation tasks.

Leveraging advancements in foundational segmentation models and language models, we integrate language prompts into the segmentation model. We then efficiently fine-tune the model using Low-Rank Adaptation (LoRA) and prompt-based tuning methods, enabling a deeper understanding of language and segmentation specific to the application domain of renal pathology.

Our proposed framework, illustrated in Fig. 3, is structured around two pivotal components: the segmentation backbone and prompt learning mechanisms. The pre-trained Efficient SAM serves as our robust segmentation backbone. Within this framework, language prompts are embedded into the Efficient SAM network to interact with spatial prompts and image tokens from the outset. Furthermore, inspired by the dynamic head concept introduced in Omniseg [13] and HATs [14], we employ embeddings of language prompts within a dynamic layer to dynamically guide and refine the segmentation process.

### A. Segmentation backbone

The Segment Anything Model (SAM) [4] is a foundational segmentation model renowned for its outstanding performance in general segmentation tasks. It utilizes the Vision Transformer (ViT) as its backbone, a structure that offers flexibility to incorporate additional prompts as input tokens to the encoder. Also, SAM has the ability to the zero-shot segmentation with points or boxes as the spatial prompts. Recently, the lightweight and efficient version, efficientSAM, was developed by distilling knowledge from the original SAM to achieve competitive performance while being much more resource-efficient. We have adopted efficientSAM [5] as our segmentation backbone in this work.

Inspired by recent advancements using dynamic heads for multi-class segmentation [15] [16], we have adapted the segmentation model to be responsive to prompts by replacing the output layers with a dynamic head controlled by prompts. The weights of the dynamic layers are derived from the latent representations learned from the concatenation of language embeddings and image features.

### B. Prompt learning

To guide segmentation effectively, our proposed pipeline model incorporates two distinct types of prompts: sparse spatial annotations and language prompts. Initially, sparse prompts like points and boxes were integrated into the SAM model for spatial annotations. However, the utilization of language prompts was previously absent in SAM.

In our approach, we leverage the pre-trained LLM model TinyLlama [17] for tokenizing and embedding free-text prompts. TinyLlama features an efficient architecture with only 1.1 billion parameters. The last token in the input sequence serves as the class token, which functions as the language prompt guiding the segmentation process. Integrating the language prompt into the segmentation network occurs through a phased approach involving three stages.

1) Add the language prompt as the additional input token to each layer of the ViT image encoder, interacting with the image tokens through self-attention.
2) Following the original structure of SAM, concatenating with the embeddings of the spatial annotation to get a prompt embedding and integrate to the mask decoder.
3) Concatenated with the image embedding to learn the weights for dynamic heads. The dynamic head will replace the upscaling layers after the vit-transform structure in the original efficient structure.

Consequently, language prompts are fused with the image segmentation backbone at different stages, enhancing the model's ability to understand and utilize segmentation-specific language.

### C. Fine-tuning

However, the pre-trained SAM's performance on medical tasks has been found to be suboptimal [18]. To address this issue, we fine-tuned the decoder of efficientSAM [5] during the training phase to enhance its performance in medical applications.

In addition to integrating a two-layer perceptron for the language embedding and combining it with the segmentation backbone, we aimed to align the language embeddings with the image embeddings by projecting them into a unified space. This adjustment was crucial for ensuring seamless integration and effective communication between the language and image modalities within our segmentation model. Furthermore, we adopted a strategy where the encoder of sparse prompts remained unfrozen. This approach has been demonstrated as effective in prior research [19], [20] on transfer learning for SAM models.

Moreover, to ensure the LLM effectively understands segmentation prompts specific to renal pathology, we employed Low-Rank Adaptation (LoRA) [21] to fine-tune the language model. The LoRA method allows us to efficiently adapt the language encoder to better suit the specific requirements of our segmentation task in the kidney pathology, enhancing its performance without significantly increasing the computational overhead.

Furthermore, in our experiments, we evaluate the model's performance on related cases that were unseen during the training phase. This evaluation aims to assess the model's flexibility and generalization capability.

## IV. DATA AND EXPERIMENTS

### A. Data

The study utilized a subset of kidney data sourced from the NEPTUNE study [22], focusing specifically on 125 patients diagnosed with minimal change disease. This subset comprised 2083 image patches, each manually annotated to identify regions such as the proximal tubule (pt), distal tubule (dt), glomerulus tuft (tuft), or glomerulus capsule (capsule), stained with Hematoxylin and Eosin (H&E) and Periodic acid-Schiff (PAS). These patches were captured at a high resolution of $1024 \times 1024$ pixels ($40 \times$ magnification, 0.25 $\mu$m). Nuclei within these patches were segmented using the trained STARdist network [23], [24] to create pseudo ground truth annotations. For detailed information on the distribution and data splits, refer to Table I.

TABLE I
SAMPLES OF PATCHES IN TRAINING, VALIDATION AND TESTING SPLIT

|  | Train | Val | Test |
|---|---|---|---|
| Distal tubule | 402 | 94 | 83 |
| Proximal tubule | 458 | 99 | 95 |
| Tuft | 287 | 62 | 67 |
| Capsule | 306 | 64 | 66 |

### B. Experiments

*1) Segmentation classes and tasks:* We developed a 4-class, 9-task segmentation dataset to evaluate the proposed methods, shown in Figure 2. Each patch includes annotations for two kidney component classes: nuclei ($c_n$) and one of **four** functional units ($c_u$) comprising proximal tubule, distal tubule, glomerulus tuft, and glomerulus capsule. We defined **nine** segmentation tasks for each patch:

1) Sgmentation of $c_u$.
2) Segmentation of $c_n$.
3) Segmentation of the union of $c_u$ and $c_n$.
4) Segmentation of the intersection of $c_u$ and $c_n$.
5) Segmentation of $c_n$ outside the $c_u$ region.
6) Segmentation of a single $c_n$ marked by a given point.
7) Segmentation of a single $c_u$ component marked by a given point.
8) Segmentation of $c_n$ within the $c_u$ marked by a given point.
9) Segmentation of $c_n$ outside the $c_u$ marked by a given point.

Images and ground-truth masks were generated for each segmentation task. Specifically, Tasks 6, 7, 8, and 9 required point inputs, with one valid point randomly generated within a unit or nucleus for each image patch. These 2D coordinates served as inputs for the spatial prompts used during segmentation.

We set these tasks because they are common and reasonable segmentation tasks in pathology image analysis, relevant yet distinct, and some tasks require both spatial prompts and textual prompts for accurate description. The model requires understanding both the logit and subject to make correct segmentation. So, these can be used to test the flexibility of the model.

To generate diverse free-text prompts for these 9 tasks, we utilized ChatGPT-4 to generate 20 different variations of each prompt, ensuring a wide range of language expressions. Each generated prompt was manually reviewed and edited to ensure linguistic diversity and clarity. As an illustrative example, the prompts used for Task 5 and Task 9 are presented in Table IV. In experiments, the first 15 prompts out of the 20 prompts were used for training the model, while the rest 5 prompts were used in the inference phase to simulate the flexibility of different expressions in language.

The Dice score was employed to evaluate the segmentation performance, comparing the results of the model with the manually annotated ground truth of units against the pseudo ground truth of nuclei.

*2) Experiments of different prompts:* To explore the influence of different prompts of tasks and units, we design three strategies to guide the segmentation. No matter which strategies are selected, the same prompt for points is held for the 4 tasks requiring points inputs.
1) **Fixed ID Embeddings:** A set of 36 embeddings (4 units $\times$ 9 tasks) was randomly initialized in a 36$\times$384 ID matrix. Each ID embedding in the shape of $[1, 384]$ corresponds to one task with one unit. These embeddings are randomly initialized and are learnable during training. Then, they are fixed during inference to guide the segmentation.
2) **Separate Fixed ID Embeddings:** Units and tasks are embedded separately, with unit embeddings in a 4$\times$384 matrix and task embeddings in a 9$\times$384 matrix. For each task on each unit, the corresponding unit embedding and task embedding are used. So, two additional tokens

are used in the vit encoder, and these two prompts are concatenated together in the integration of the other two stages mentioned in the method section. Same as the above setting, these embeddings are learnable during training and fixed during inference to guide the segmentation.
3) **Free Text Embeddings from LLM:** For each task, we designed 20 different prompts, with the names of units filled correspondingly. The embeddings of these language prompts are generated from the TinyLlama-1.1B-Chat model [17]. The last token in the output, typically used as the class token for classification [1], was used as the language prompt to integrate with the segmentation backbone. The dim of the text token was [1,1024], and it was projected to [1,384] to be comparable with the image tokens through two linear layers.

*3) Experiments of complete and incomplete training set:* To explore the model's generalization ability to unseen cases, we removed all tasks involving distal tubule and glomerulus capsule data from the training set, except for task 1 (segmentation of units). Task 1 is saved because the concept of these classes should be introduced. This approach allows us to assess whether learning the basic concept of new classes enables the model to apply its segmentation abilities to other tasks for these classes.

In the evaluation, we remove the evaluation results of tasks 1, 2 and 6 which are available in the training set for all 4 classes of units, the average dice score of the other 6 tasks are used to compare and showed in the experiment results in Table III.

*4) Experiments of fintuning:* We also conducted an ablation study comparing the performance of a frozen language encoder with that of a language encoder fine-tuned using the Low-Rank Adaptation (LoRA) method. In our experiments, the LoRA configuration was specifically set to rank = 8, alpha = 16, and dropout = 0.05. This setup resulted in 1,126,400 trainable parameters, which constituted only 0.1% of the total model weights.

*5) Other settings and parameters:* For the segmentation backbone, we used the efficientSAM [2] with the dynamic head part from [3]. The dynamic network replaced the upscale part of the efficient SAM to get the final segmentation with two channels. And we use the TinyLlama-1.1B-Chat [4] as the text encoder.

Same as the efficientSAM and SAM, the image in resolution 1024$\times$1024 is used in the input.

For weight optimization, we employed the Adam optimizer starting from the learning rate of 0.001, with a decay factor of 0.99. All experiments were trained for 900 epochs. 1000 samples were randomly selected from the training set for each epoch, with the batch size equals to 1. Model performance was evaluated using the validation set every 100 epochs, and

[1] https://huggingface.co/docs/transformers/en/model_doc/llama
[2] https://github.com/yformer/EfficientSAM
[3] https://github.com/ddrrnn123/Omni-Seg
[4] https://huggingface.co/TinyLlama/TinyLlama-1.1B-Chat-v1.0

TABLE II
PERFORMANCE OF SEGMENTATION RESULTS WHEN USING DIFFERENT
PROMPTS AND COMPLETE/INCOMPLETE TRAINING SET. THE BEST
PERFORMANCE OF THE DICE SCORE IS HIGHLIGHTED IN **BOLD**, WHILE
THE SECOND-BEST PERFORMANCE IS HIGHL1GHTED BY UNDERLINE.

| Prompts | Complete Training Set | | | |
|---|---|---|---|---|
| | dt | capsule | pt | tuft |
| One-set Fixed ID | 0.6728* | 0.7646 | 0.6661* | 0.7585 |
| Two-set Fixed ID | **0.6777*** | **0.7814** | **0.6741** | **0.7847** |
| Freetext | 0.6690 | 0.7246 | 0.6524 | 0.7230 |
| | Incomplete Training Set | | | |
| | Unseen | | Seen | |
| Prompts | dt | capsule | pt | tuft |
| One-set Fixed ID | 0.2200 | 0.1325 | 0.6590* | 0.7516 |
| Two-set Fixed ID | **0.4630** | **0.6470** | **0.6798** | **0.7700** |
| Freetext | 0.3769 | 0.5672 | 0.6423 | 0.7090 |

*There is no significant difference between the results marked with "*" and the corresponding results of freetext, with a p-value of Wilcoxon t-test >= 0.05.

TABLE III
ABLATION STUDY TO EVALUATE THE IMPACT OF 1) FINE-TUNING THE
TEXT ENCODER AND 2) INCLUDING FREE-TEXT PROMPTS FROM THE
INFERENCE PHASE IN THE TRAINING PHASE. THE BEST PERFORMANCE IS
HIGHLIGHTED IN **BOLD**, WHILE THE SECOND-BEST PERFORMANCE IS
HIGHLIGHTED BY UNDERLINE.

| Train/test text | Text Encoder | Complete Training Set | | | |
|---|---|---|---|---|---|
| | | dt | capsule | pt | tuft |
| Different | Freeze | 0.4439 | 0.4603 | 0.4879 | 0.5185 |
| Same | LoRA | **0.6888** | **0.7749** | **0.6870** | **0.7738** |
| Different | LoRA | 0.6690 | 0.7246 | 0.6524 | 0.7230 |

*There is a significant difference between each pair of experiments, with a p-value of Wilcoxon t-test < 0.05.

the average Dice score was calculated to identify the optimal epoch for testing. The average Dice score served as the evaluation metric. All experiments were conducted on an NVIDIA RTX A6000 GPU, while the experiment of finetuning took up about 7GB of GPU memory.

## V. RESULTS AND DISCUSSION

In Table II, we compare the impact of different prompt settings on segmentation results using either a complete or incomplete training set. A complete training set includes all nine tasks corresponding to the four kidney unit classes during the training phase, while an incomplete training set means that unseen classes, such as dt and capsule, only include the basic tasks 1 and 2 during training and exclude other tasks. This setup is designed to test the model's generalizability to unseen cases.

The experimental results show that with a complete training set, fixed task IDs consistently outperform free-text prompts. This outcome is expected because fixed task IDs provide a more stable and consistent guide for the model, whereas free-text prompts introduce variability. The language descriptions used during training and testing with free-text prompts are not entirely consistent, making it more challenging to guide the model effectively.

However, it can be seen that with an incomplete training set, free-text prompts perform better on unseen tasks compared to the less flexible single set fixed IDs with significant differences. In this scenario, the two-set fixed ID performs better because fixed IDs can be seen as precise definitions of tasks and unit classes, providing a clear and accurate guide for the model. Free-text prompts, while offering greater flexibility in expression, sacrifice some accuracy due to their variability. The accuracy difference is not significant compared with the prediction of dt and pt using one-set fixed ID with p-value from the Wilcoxon t-test larger than 0.05.

In Table III, we conducted an ablation study to illustrate the positive impact of fine-tuning the text encoder on segmentation results. Also, the results show that the model's performance approaches that of fixed IDs when the free-text instructions used during the testing phase are also included in the training phase. When the free-text instructions in the testing phase are not included in the training phase, the model's performance slightly decreases but is still much better than the frozen one, indicating the language model's ability to generalize to different expressions of the same task.

Experiments demonstrate that using combined prompts or descriptive free text as prompts to guide kidney segmentation tasks enables the model to generalize to unseen cases. Furthermore, when using free-text prompts, the model can achieve results close to those obtained with previously seen language expressions for the same content, even when encountering new and unseen expressions. This highlights the model's flexibility and generalizability in terms of both language expression and content.

However, this experiment also has some limitations. Due to limited experimental data and scenarios, the experiments were conducted on a relatively small scale and in simulated conditions. This setup provides only preliminary validation of the feasibility of using language-guided fine-grained segmentation in medical imaging. For practical applications in more realistic settings, further verification with greater computational power and data is needed.

In future research, 1) we plan to enhance model flexibility and generalization by increasing both prompt and task diversity. First, expand training data and improve language flexibility by simulating or collecting more varied language prompts. Second, diversify and refine the segmentation task scope, such as segmenting a specific number of cells in an image, to increase task diversity. 2) Furthermore, evaluating model performance in more practical clinical environments will involve assessing zero-shot and few-shot capabilities using additional datasets, and introducing human-in-the-loop approaches for evaluation and interactive segmentation. Revisions from pathologists can be used as visual prompts to continue refining segmentation results. 3) Additionally, Explore advanced models and training strategies, including alternative vision-language fusion architectures for more effective image-language fusion and more recent foundational language models for advanced language understanding. The above efforts are aimed at advancing the robustness and applicability of language-guided segmentation methods for medical images, ultimately aiming to improve diagnostic accuracy and clinical

TABLE IV
LANGUAGE PROMPTS FOR SEGMENTATION. THE <CLASS> IN PROMPTS IS REPLACED BY THE NAME OF THE UNIT CLASSES IN BOTH THE TRAINING AND INFERENCE. (ROW 1-15 WERE USED FOR TRAINING AND THE LAST 5 WERE USED FOR INFERENCE). TAKE TASK NO.9 AND TASK NO.5 AS EXAMPLES.

| Index | Prompt |
|---|---|
| | Task 9 |
| 1 | At the designated point, segment the nuclei found outside of the <class> that is centered at the point. |
| 2 | Outside of the region <class> marked by the provided point, segment all nuclei. |
| 3 | Segment nuclei that are located out of the <class> which was pointed at the provided point. |
| 4 | There are nuclei located out of the <class> that are pointed by the provided point. Segment these nuclei out. |
| 5 | Perform segmentation on nuclei located beyond the boundaries of the <class> that is marked by the point provided. |
| 6 | Accurately delineate the boundaries of the nuclei outside of the <class> that is located at the given point. |
| 7 | Mark the boundaries of nuclei falling out of the specified <class> that is at the given location. |
| 8 | There is a <class> at the given point. Segment every nucleus outside of that <class>. |
| 9 | Given a point, segment the nuclei located out of the region of the <class> that is pointed by the provided point. |
| 10 | There is a provided point. Outline all nuclei located out of the <class> which is pointed at the provided point. |
| 11 | Outline and segment the nuclei situated in the <class> that is identified by the point shown. |
| 12 | Proceed to segment the nuclei found out of the marked <class>. |
| 13 | Identify and delineate the boundary of each nucleus outside of the pointed-at <class>. |
| 14 | Encircle and segment each nucleus exterior to the specific <class> that is highlighted at the indicated point. |
| 15 | For the nuclei located outside of the <class> that is centered at the pointed spot, perform segmentation. |
| 16 | Locate and segment every nucleus out of the <class> marked by the given point. |
| 17 | Trace and segment nuclei exterior to the region of the pinpointed <class>, defining precise boundaries of these nuclei. |
| 18 | Define the edges of nuclei that are shown outside of the area of a pointed <class>. |
| 19 | Locating a <class> at the given point, only segments the nuclei beyond the boundary of that <class>. |
| 20 | Outside the pointed <class>, outline the nuclei for segmentation. |
| | Task 5 |
| 1 | Outline nuclei situated within <class> in this image. |
| 2 | Segment nuclei inside <class> in this image. |
| 3 | Perform segmentation on all nuclei within <class> in this image. |
| 4 | Detect and segment every nucleus located inside the <class> in this image. |
| 5 | There are nuclei inside of the <class> region. Segment these nuclei out. |
| 6 | Delineate the boundaries of nuclei within the area of <class>. |
| 7 | Nuclei inside of the <class> region need to be segmented out. |
| 8 | Identify and segment nuclei located within <class> in this image. |
| 9 | Extract and outline all nuclei falling inside of <class> in this image. |
| 10 | Proceed to segment all nuclei found inside the boundary of <class>. |
| 11 | Accurately segment the boundary of every nucleus belonging to the <class> area. |
| 12 | Delineate the edges of nuclei inside of all <class> in this image. |
| 13 | Delineate the intersection of <class> region and nuclei in this image. |
| 14 | Display the overlapping area between the nuclei and <class> in this image. |
| 15 | Segment all nuclei contained in <class>. |
| 16 | Segment all nuclei within the boundary of <class> in this image. |
| 17 | Every nucleus located within the <class> is expected to segment. |
| 18 | Perform segmentation on all nuclei inside of <class> area in this image. |
| 19 | Locate and outline all nuclei inside of <class> in this image. |
| 20 | Output the intersection of the nuclei and <class> region in this image. |

outcomes in medical practice.

## VI. CONCLUSION

In this work, we designed a language-guided pathology image segmentation model and conducted experiments on a renal pathology dataset for multiclass and multitask segmentation, followed by quantitative evaluation. Our experiments showed that language-guided segmentation offers greater diversity and flexibility compared to unique and fixed task encoding, effectively handling unseen cases. By using LoRA for fine-tuning, we ensured the model effectively understands segmentation prompts specific to renal pathology.

Due to limited data and computational resources, we did not conduct large-scale experiments. However, our preliminary exploration has shown promising results, indicating the model's effectiveness and potential for future development.

## ACKNOWLEDGMENT

This research was supported by NIH R01DK135597 (Huo), NSF CAREER 1452485, NSF 2040462, NCRR Grant UL1-01 (now at NCATS Grant 2 UL1 TR000445-06), resources of ACCRE at Vanderbilt University.

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
