# OpenReview forum: "Enhancing Physician Flexibility: Prompt-Guided Multi-class Pathological Segmentation for Diverse Outcomes"
_IEEE.org/EMBS/BHI/2024/Conference — IEEE BHI'24_

### Official Review · Reviewer_nFTc · 2024-08-02
**Use of Large Language Models for Flexible Multi-Class Medical Image Segmentation.**

**Overall Rating:** 7
**Confidence:** 5

**Other Quality Metrics:**

(a) Clarity of writing; Great
 (b) Clinical Significance; Great
 (c) Methodological Novelty; Good
(d) Experiments and Results: Good

**Questions For The Authors:**

Firstly, congratulations on your nice work. Your innovative approach has the potential to significantly advance medical image segmentation.

Testing on Diverse Datasets: Have you considered evaluating your model on a wider range of medical datasets representing different pathological conditions and imaging modalities? This could provide a more comprehensive understanding of your model's generalizability and robustness.

Real-World Implementation Challenges: Could you elaborate on the potential challenges and considerations for deploying your model in real-world clinical settings? Specifically, how do you foresee addressing issues related to computational resource requirements, integration with existing healthcare systems, and training medical professionals to effectively use the model?

**Strengths:**

Innovative Approach: The use of Large Language Models (LLMs) to enhance segmentation in medical image analysis is a novel idea, introducing a new dimension to AI deployment in clinical settings.
Real-World Applicability: The proposed method shows great potential to be implemented in real-life clinical scenarios, potentially improving the accuracy and efficiency of pathological image segmentation.
Thorough Evaluation: The authors have conducted comprehensive testing and comparison with existing methods, demonstrating the advantages of their approach. They have successfully addressed several challenges inherent in current systems.

**Summary Of The Paper:**

1. The paper introduces an innovative approach to enhancing medical image segmentation by integrating Large Language Models (LLMs) to guide flexible multi-class segmentation tasks.
2. The authors have constructed a computationally efficient pipeline that leverages fine-tuned language prompts to improve segmentation accuracy and flexibility, especially in complex scenarios such as kidney pathology.
3. Their methodology is thoroughly evaluated against traditional task tokens, showcasing significant improvements in adaptability and performance.
4. The paper presents a multi-task kidney pathology dataset with various free-text prompts and demonstrates the model's ability to handle new cases during inference, highlighting its potential in real-world clinical applications.
5. The approach addresses critical challenges in the field, offering valuable insights and a promising direction for future research in medical image analysis.

**Weaknesses:**

Diverse Dataset Testing: While the authors have demonstrated the model's effectiveness on a multi-task kidney pathology dataset, the work would benefit from testing on a more diverse set of medical images. Evaluating the model on various datasets representing different pathological conditions and imaging modalities would provide a more comprehensive assessment of its generalizability and robustness.
Real-World Deployment Challenges: The paper could address potential challenges related to the deployment of such models in real-world clinical settings. These include considerations like computational resource requirements, integration with existing healthcare systems, and user training for medical professionals.
Quantitative Analysis: Although the authors have compared their model with existing methods, a more detailed quantitative analysis, including statistical significance tests, would strengthen the validity of their claims regarding performance improvements.

---

### Official Review · Reviewer_MyiA · 2024-08-14
**High-level publication in the field of prompt based segmentation**

**Overall Rating:** 8
**Confidence:** 3

**Other Quality Metrics:**

(a) Clarity of writing - excellent
(b) Clinical Significance - great
(c) Methodological Novelty - great
(d) Experiments and Results - great

**Questions For The Authors:**

STOTA:
- The related Work section could be combined with the Introduction

Methods:
- Did you already evaluate a feedback loop in the training process? Let the physicians rate the resulting segmentation to optimize the performance.

Results and Discussion:
- Even if the prompts are longer than simple "one word" classifications/IDs, in my opinion, the prompts given in Table II are language-wise very similar. Could the network get confused when the prompts are not distinguishable?
- Could you also comment on how far prompt-based segmentation could create an observer bias? Different physicians or clinics might use different descriptions for the same output.

**Strengths:**

- very good visualizations and clear textual descriptions that emphasize the proposed methods and problems
- a methodologically sound approach that combines recent methods in a professional way
- no formal errors, professional manuscript

**Summary Of The Paper:**

The authors present a pathological segmentation approach based on flexible prompts and spatial information. Compared to fixed ID-based classifications, the network is trained to infer the segmentation task from versatile prompts. This promises a more flexible segmentation assistant.

**Weaknesses:**

- freetext prompt results do not show the best DICE scores - Why?
- discussion could emphasize the next necessary steps in research more clearly

---

### Decision · Program_Chairs · 2024-09-23

Accept